**PLOS** NEGLECTED TROPICAL DISEASES

# The disability-adjusted life years (DALYs), prevalence and incidence of scabies, 1990–2021: A systematic analysis from the Global Burden of Disease Study 2021

**Jiajia Li**, **Zehu Liu** *, **Xiujiao Xia** *

Department of Dermatology, Hangzhou Third People's Hospital, Hangzhou Third Hospital Affiliated to Zhejiang Chinese Medical University, Hangzhou, Zhejiang Province, China

* zehuliu@yahoo.com (ZL); 804534095@qq.com (XX)

**Data Availability Statement:** All relevant data are included in the manuscript and its Supporting Information files. Additionally, all raw data are

## Abstract

### Background

Current literature lacks a recent global analysis of scabies. This study aims to analyze the burden and epidemiological characteristics of scabies using data from the Global Burden of Disease (GBD) 2021 study.

### Methodology/Principal findings

The analysis assessed disability-adjusted life years (DALYs), prevalence, and incidence of scabies from 1990 to 2021, stratified by geographic location, socio demographic index (SDI), gender, and age. In 2021, scabies caused 5.3 million DALYs, 206.6 million prevalence, and 622.5 million incidence, primarily affecting children and young people. The burden was heaviest in middle SDI regions and lowest in high SDI regions. Oceania, Tropical Latin America, and East Asia ranked as the top three regions in global scabies burden. Nationally, Fiji, Guam, Tonga, Tuvalu, and Northern Mariana Islands had the highest age-standardised DALY rates. From 1990 to 2021, global age-standardized rates (ASRs) of DALYs, prevalence, and incidence for scabies declined, while the absolute numbers increased. These ASRs showed an upward trend in high and high-middle SDI regions, with significant increases in Central Latin America and high-income North America. Larger burden increases were observed in Sri Lanka, the United States of America, and Mexico compared to other countries and territories. In terms of age, these ASRs increased from 40, particularly for women and the elderly.

### Conclusions/Significance

The global scabies burden was higher in tropical regions, particularly among children and young people, in 2021. Between 1990 and 2021, the burden increased in higher SDI regions, Central Latin America, and high-income North America, warranting focused attention. Additionally, the rising burden among adults over 40, particularly women and the elderly, highlighted the need for targeted interventions.

publicly available on Figshare at https://doi.org/10.6084/m9.figshare.27987941.v1.

**Funding:** This work was supported by the Hangzhou Science and Technology Bureau, China (grant no. 202004A17 to ZL) and the Hangzhou Health Science and Technology Plan, China (grant no. Z2024015 to ZL). The funders had no role in study design, data collection and analysis, decision to publish, or preparation of the manuscript.

**Competing interests:** The authors report there are no competing interests to declare.

## Author summary

Scabies is an infectious disease predominantly prevalent in tropical and humid regions, particularly affecting children and the elderly. The disease is caused by the infestation of *Sarcoptes scabiei* mite and transmitted through direct skin contact. Risk factors for this disease include overcrowding, poverty, and unclean water sources. The primary symptom is severe itching, which is more pronounced at night and can lead to insomnia. Scabies can significantly impact the quality of life and incur substantial treatment costs, making it a public health issue of considerable concern. However, research on scabies remains insufficient, and its patterns of prevalence and incidence require further exploration. The GBD study offers free, continuously updated data on hundreds of diseases, including scabies, providing scientific and reliable results. Using the latest high-quality GBD 2021 data, this study examined global scabies prevalence, incidence, and DALYs across different regions, genders, and age groups. Findings indicated that the burden remained high in tropical regions, children, and adolescents, despite some decline. Moreover, burden of scabies has increased in higher SDI regions, and individuals over 40 years old, particularly women and the elderly. This analysis provided a comprehensive overview of global scabies burden, revealing its recent trends and may be helpful in the formulation of scabies control policies.

## Introduction

Scabies, a skin condition caused by infestation of the parasitic mite *Sarcoptes scabiei*, manifests clinically with pruritus, burrows, papules, and erythematous patches [1]. When the skin barrier is compromised, it often leads to secondary bacterial skin infections. Transmission of scabies primarily occurs through direct skin-to-skin contact, with occasional indirect transmission via contaminated items, the latter being more common in crusted scabies with higher mite burdens [2]. Scabies is more prevalent in resource-limited, hot, and humid tropical regions. In recent years, scabies has become more widespread, with risk factors including poverty, overcrowding, lack of clean water sources, and large-scale migration [3]. According to the 2020 consensus criteria established by the International Alliance for the Control of Scabies (IACS), three diagnostic levels were defined: confirmed scabies, clinical scabies, and suspected scabies. The diagnosis of confirmed scabies can be made through identification of mites, eggs, or feces [4]. While scabies remains treatable with medications such as ivermectin, permethrin, benzyl benzoate, and crotamiton, reports of reduced in vitro susceptibility and clinical resistance to the primary scabicides, permethrin and ivermectin, have emerged. These observations, coupled with concerns about treatment efficacy in recent years, underscore the need for further investigation into the underlying causes of potential resistance and treatment failure [5]. Additionally, during treatment, it is essential that cohabiting family members are also treated and the surrounding environment is properly disinfected [6].

The results of the Global Burden of Disease (GBD) study serve as a scientific, reliable, publicly accessible data resource contributing significantly to global health. Previous studies have analysed the burden and trends of scabies using GBD 2015 and GBD 2017 data. Dr. Karimkhani *et al.* analysed scabies disability-adjusted life years (DALYs) using GBD 2015 data, revealing a greater burden in tropical regions, with the top five countries bearing the heaviest burden being Indonesia, China, Timor-Leste, Vanuatu, and Fiji. They also found that children, adolescents, and the elderly bore the greatest DALY burden [7]. Dr. Zhang *et al.* examined scabies prevalence and incidence using GBD 2017 data, noting increases in both indicators in

high sociodemographic index (SDI) regions, high-income North America, and individuals over 70 years old [8]. As early as 1689, descriptions of scabies existed [9]. Over time, the prevalence of scabies has gradually changed in various parts of the world, attracting increased attention due to the significant health and economic burdens it imposes [8,10]. Despite advancements, the regional characteristics, gender, and age disparities in scabies prevalence remain inadequately understood and continually evolving. Effective prevention and control of scabies continue to be challenging issues worldwide, requiring ongoing research efforts to deepen our understanding of the disease.

GBD 2021 updated the burden of 371 diseases and injuries, including scabies, across 204 countries and territories, building upon GBD 2019 with various improvements [11]. Our goal is to analyse burden of scabies using high-quality data from GBD 2021, describing the scabies DALYs, prevalence, and incidence by geography, gender, and age in 2021, as well as the trends from 1990 to 2021. This analysis may facilitate the development of region-specific scabies prevention and control measures, providing more protection for those severely affected or at risk of scabies.

## Methods

### Overall

All data resources were obtained from the GBD Results Tool (http://ghdx.healthdata.org/gbd-results-tool) of Global Health Data Exchange (GHDX) platform, which was maintained by the institute for Health Metrics and Evaluation (IHME).

The methodology for GBD study has been extensively described elsewhere [7,11,12]. The GBD 2021 represents a new iteration updating the estimates from previous rounds of the GBD study. Scabies was categorized within the GBD 2021 cause group of skin and subcutaneous conditions and was assigned the disability weight for disfigurement level 1 with itch/pain. The lay description for this level states, "The individual has a slight, visible physical deformity that is sometimes sore or itchy. Others notice the deformity, which causes some worry and discomfort." Disability weights, ranging from 0 (no health loss) to 1 (equivalent to death), represent health loss from a sequela. The disability weight assigned to scabies was 0.027, which was used to calculate the Years Lived with Disability (YLDs). YLDs quantify non-fatal health loss, years of life lost (YLLs) quantify fatal health loss. DALYs are calculated by summing YLDs and YLLs, quantifying both years lost to premature mortality and years lived with disability, enabling comparisons across diseases. For scabies, DALYs were calculated assuming YLLs were zero, making YLDs equivalent to DALYs. According to the International Classification of Diseases (ICD), scabies was coded as ICD-10: B86. [11].

In our analysis, DALYs, prevalence, and incidence attributed to scabies were examined, utilizing both number and rate metrics.

### Age, sex, time period and geographical units

For GBD 2021 study, scabies prevalence across different demographics, including sex, age, year, and geographical location, were estimated utilizing DisMod-MR 2.1, a Bayesian meta-regression modeling tool. The consistency of incidence and prevalence data was ensured also through the DisMod-MR 2.1. Country-level covariates including sociodemographic index, sugar consumption, and the Healthcare Access and Quality index were employed to guide estimates of scabies for countries with limited or absent data. The summary exposure value (SEV) for unsafe water was used as only location-level covariate, with a minimum coefficient of variation set at 0.4 [11].

In GBD 2021, all estimates were disaggregated into 25 age groups, ranging from 0–6 days to 95+ years. Compared to GBD 2019, GBD 2021 added age groups 1–5 months, 6–11 months, and 12–23 months. Moreover, estimates were calculated globally, covering seven super-regions, 21 regions, 204 countries and territories, and 811 subnational locations. Nine countries and territories, including Cook Islands, Monaco, Nauru, Niue, Palau, San Marino, Saint Kitts and Nevis, Tokelau, and Tuvalu, were added since GBD 2019. Based on socio-economic factors associated with total fertility rate, mean education, and lag-distributed income (LDI) per capita, the 204 countries and territories were classified into five SDI regions (low, low-middle, middle, high-middle, high), with the SDI values in GBD 2021 scaled by a factor of 100, ranging from 0–100 [11,12].

In our study, burden of scabies was examined across males, females, and both genders combined. All 25 age groups were analysed. Geographically, the analysis included the globe, five SDI regions, 21 regions, and 204 countries and territories. Temporally, the burden of scabies in 2021 and its trends from 1990 to 2021 were assessed. Percentage change demonstrated the overall increase or decrease in disease burden from 1990 to 2021 by comparing values at the start and end of the period. The estimated annual percentage changes (EAPCs) in age-standardized rate (ASR [per 100000 population]) were calculated to quantify the average annual trend, accounting for year-to-year variations over time [13]. A linear relationship is assumed between ln (ASR) and year, expressed as $Y = \alpha + \beta x + \varepsilon$, where Y = ln (ASR), X = calendar year, β represents the regression coefficient and ε is the error term (residual). The EAPC was calculated using the formula $100 \times (exp(\beta) - 1)$. If both the EAPC estimate and the lower boundary of its 95% CI were positive, an increasing trend in ASR was indicated. Conversely, if both the EAPC estimate and the upper boundary of its 95% CI were negative, a decreasing trend in ASR was indicated.

## Statistical analysis

Data analysis and visualization were conducted using R version 4.3.2, a programming software for statistical computing and graphics. Ethical approval was not required as all data utilized in this study were sourced from anonymized public databases.

## Results

### Burden of scabies in various geographical locations

**Global level.**   In 2021, scabies resulted in 5.3 million (95% UI 3.0–8.7) DALYs, with contributions of 2.7 million (1.5–4.5) from males and 2.6 million (1.5–4.3) from females. Among 206.6 million (184.2–231.7) prevalence, males comprised 105.5 million (94.0–118.4) and females 101.0 million (90.1–118.4). Additionally, there were 622.5 million (556.2–695.0) incidence, including 318.1 million (284.0–355.1) males and 304.4 million (271.9–339.6) females. The scabies ASRs of DALYs, prevalence and incidence were 68.7 (38.3–113.3) per 100000 people, 2666.5 (2368.2–2994.5), 8049.5 (7165.2–9024.2), respectively. Between 1990 and 2021, the numbers of DALYs, prevalence and incidence increased, while their corresponding ASRs declined, with a more pronounced decrease observed among males. (Tables 1–3 and S1 Fig).

**Regional level.**   In 2021, the burden of scabies was highest in middle SDI regions and lowest in high SDI regions. The ASRs of DALYs, prevalence and incidence in middle SDI regions were 97.3 (54.4–160.0), 3774.7 (3361.3–4223.8), 11409.1 (10174.5–12796.2), respectively. From 1990 to 2021, the ASRs of DALYs, prevalence and incidence showed an upward trend in high and high-middle SDI regions, with the most significant increases in high SDI regions. Conversely, the ASRs of DALY, prevalence and incidence declined in middle, low-middle, and low SDI regions.

**Table 1. Scabies DALYs in 2021 and trends from 1990 to 2021.**

| Characteristics | Number (thousands) in 2021 (95% UI) | Percentage change (%) in number, 1990–2021 (95% UI) | ASR (per 100000) in 2021 (95% UI) | Percentage change (%) in ASR, 1990–2021 (95% UI) | EAPC of ASR (95% CI) |
|---|---|---|---|---|---|
| | | | DALYs | | |
| Global | 5315.5 (2968.4 to 8740.3) | 34.3 (30.6 to 38.4) | 68.7 (38.3 to 113.3) | -2.6 (-3.5 to -1.7) | -0.03 (-0.03 to -0.03) |
| Sex | | | | | |
| Male | 2727.1 (1520.0 to 4494.9) | 33.4 (30.0 to 37.5) | 69.8 (38.9 to 115.2) | -3.2 (-4.1 to -2.2) | -0.05 (-0.06 to -0.05) |
| Female | 2588.5 (1451.1 to 4245.6) | 35.3 (31.4 to 39.6) | 67.6 (37.8 to 111.3) | -2.1 (-3.2 to -1.1) | -0.01 (-0.01 to -0.02) |
| SDI | | | | | |
| High SDI | 151.9 (86.1 to 243.7) | 31.7 (26.2 to 37.9) | 14.9 (8.4 to 24.2) | 8.7 (6.0 to 11.7) | 0.54 (0.59 to 0.49) |
| High-middle SDI | 862.0 (486.5 to 1399.7) | 20.8 (15.2 to 27.4) | 71.1 (39.8 to 117.2) | 7.1 (5.5 to 8.8) | 0.18 (0.18 to 0.17) |
| Middle SDI | 2294.4 (1287.9 to 3754.0) | 23.7 (19.6 to 28.8) | 97.3 (54.4 to 160.0) | -5.1 (-5.8 to -4.3) | -0.09 (-0.10 to -0.09) |
| Low-middle SDI | 1424.2 (791.0 to 2354.8) | 44.9 (40.0 to 49.0) | 72.4 (40.2 to 119.2) | -5.8 (-6.7 to -5.0) | -0.08 (-0.09 to -0.07) |
| Low SDI | 578.2 (319.6 to 959.5) | 100.9 (97.0 to 104.7) | 48.1 (26.8 to 79.0) | -7.8 (-9.0 to -6.7) | -0.26 (-0.28 to -0.25) |
| Region | | | | | |
| Andean Latin America | 62.8 (34.8 to 101.7) | 57.8 (51.3 to 64.8) | 95.8 (53.2 to 154.9) | 0.2 (-2.1 to 2.3) | 0.01 (0.01 to 0.01) |
| Australasia | 0.9 (0.5 to 1.4) | 41.6 (29.7 to 55.5) | 3.3 (1.8 to 5.3) | 0.5 (-7.7 to 9.7) | 0.32 (0.35 to 0.28) |
| Caribbean | 43.7 (24.3 to 71.2) | 24.5 (20.7 to 28.7) | 95.7 (53.1 to 155.6) | 0.1 (-1.5 to 1.6) | -0.00 (0.00 to 0.01) |
| Central Asia | 14.1 (7.8 to 23.1) | 30.7 (26.1 to 36.1) | 14.8 (8.2 to 24.5) | 0.1 (-2.4 to 2.8) | 0.00 (-0.00 to 0.02) |
| Central Europe | 40.0 (22.5 to 64.7) | -12.2 (-15.0 to -9.1) | 37.8 (21.1 to 62.0) | 1.1 (-0.3 to 2.4) | 0.05 (0.08 to 0.04) |
| Central Latin America | 210.5 (116.8 to 345.8) | 47.7 (42.2 to 53.8) | 85.2 (47.3 to 140.0) | 9.1 (7.6 to 10.5) | 0.55 (0.55 to 0.52) |
| Central Sub-Saharan Africa | 35.8 (19.7 to 58.5) | 151.0 (141.2 to 160.9) | 24.9 (13.8 to 40.2) | 1.0 (-1.9 to 4.1) | 0.03 (0.01 to 0.04) |
| East Asia | 1692.2 (957.2 to 2749.2) | 8.7 (3.2 to 15.2) | 124.4 (69.7 to 205.5) | 0.2 (-0.8 to 1.3) | 0.00 (0.00 to 0.00) |
| Eastern Europe | 14.3 (8.0 to 23.2) | -13.2 (-15.9 to -10.2) | 7.9 (4.4 to 13.0) | 3.4 (1.8 to 5.3) | 0.12 (0.12 to 0.12) |
| Eastern Sub-Saharan Africa | 215.6 (119.4 to 356.5) | 100.1 (92.7 to 106.6) | 46.7 (26.1 to 76.5) | -9.0 (-11.6 to -6.6) | -0.65 (-0.67 to -0.64) |
| High-income Asia Pacific | 3.2 (1.8 to 5.1) | -5.7 (-11.9 to 1.1) | 2.0 (1.1 to 3.4) | 1.0 (-3.6 to 6.0) | 0.02 (-0.01 to 0.06) |
| High-income North America | 11.4 (6.3 to 18.5) | 12.6 (6.3 to 19.1) | 3.4 (1.9 to 5.5) | -7.0 (-12.2 to -1.4) | 1.12 (1.17 to 1.02) |
| North Africa and Middle East | 163.8 (90.4 to 271.1) | 64.2 (58.9 to 70.3) | 26.0 (14.3 to 43.0) | -3.6 (-6.2 to -1.1) | -0.14 (-0.19 to -0.12) |
| Oceania | 29.7 (16.6 to 49.3) | 106.2 (99.9 to 111.5) | 203.2 (114.3 to 336.3) | -0.7 (-2.8 to 1.5) | -0.04 (-0.04 to -0.02) |
| South Asia | 1481.6 (822.2 to 2454.4) | 55.8 (50.3 to 60.4) | 78.9 (43.8 to 130.4) | 0.2 (-0.6 to 0.9) | 0.24 (0.24 to 0.24) |
| Southeast Asia | 822.4 (460.6 to 1347.7) | 36.4 (31.8 to 41.9) | 121.2 (67.6 to 198.9) | 0.4 (-0.5 to 1.3) | 0.07 (0.07 to 0.08) |

(*Continued*)

**Table 1.** (Continued)

| Characteristics | DALYs | | | | |
| --- | --- | --- | --- | --- | --- |
| | Number (thousands) in 2021 (95% UI) | Percentage change (%) in number, 1990–2021 (95% UI) | ASR (per 100000) in 2021 (95% UI) | Percentage change (%) in ASR, 1990–2021 (95% UI) | EAPC of ASR (95% CI) |
| Southern Latin America | 1.3 (0.7 to 2.1) | 29.1 (16.2 to 43.4) | 2.0 (1.1 to 3.2) | 0.9 (-9.1 to 11.9) | 0.02 (0.04 to 0.02) |
| Southern Sub-Saharan Africa | 20.9 (11.7 to 34.0) | 45.6 (41.3 to 50.7) | 25.6 (14.2 to 41.6) | -0.3 (-1.9 to 1.3) | -0.01 (-0.02 to -0.02) |
| Tropical Latin America | 327.4 (183.4 to 528.5) | 35.2 (29.6 to 42.0) | 153.0 (85.5 to 247.8) | 0.2 (-0.7 to 1.2) | -0.11 (-0.13 to -0.12) |
| Western Europe | 1.4 (0.8 to 2.3) | 5.2 (-0.6 to 11.5) | 0.4 (0.2 to 0.6) | 0.1 (-5.1 to 5.8) | 0.11 (0.10 to 0.08) |
| Western Sub-Saharan Africa | 122.5 (69.2 to 202.4) | 154.7 (151.5 to 157.6) | 23.8 (13.5 to 38.7) | -0.3 (-1.3 to 0.7) | -0.03 (-0.04 to -0.00) |

In 2021, East Asia, South Asia, and Southeast Asia emerged as the top three world regions with the highest numbers of DALYs, prevalence, and incidence. However, the regions ranked highest age-standardised DALY, prevalence, and incidence rates were Oceania (203.2 [114.3–336.3], 7899.3 [7022.8–8880.3], 23783.1 [21294.9–26555.7]), Tropical Latin America (153.0 [85.5–247.8], 5963.1 [5310.6–6767.3], 17921.1 [16005.4–20289.7]), and East Asia (124.4 [69.7–205.5], 4806.8 [4279.4–5402.3], 14562.3 [12977.0–16303.0]). Despite declines in ASRs of DALYs, prevalence, and incidence in some regions between 1990 and 2021, many other regions showed increases. Notably, the growth was more pronounced in Central Latin America (percentage change: 9.1% [7.6–10.5]) and High-income North America (EAPC: 1.12 [1.17 to 1.02]). Additionally, the ASRs of prevalence and incidence in these three regions also exhibited significant increases (Tables 1–3 and S1 Fig).

**National scale.** As showed in Fig 1, among 204 countries and territories, the top ten countries with largest numbers of scabies DALYs, prevalence and incidence were the same in 2021, and were arranged in the same order. China led the first, followed by India, Indonesia, Brazil, Pakistan, Philippines, Bangladesh, Vietnam, Mexico, and Nigeria. In China, the numbers of DALYs, prevalence and incidence were 1.6 million (0.9–2.7), 63.7 million (56.9–70.9), 192.1 million (171.5–213.8), respectively.

In 2021, ASR of DALYs ranged from 225.8 per 100000 people [129.0–369.0] in Fiji to 0.1 (0.1–0.2) in Germany. ASR of prevalence varied from 8769.4 (7754.1–9833.9) in Fiji to 5.2 (4.5–6.0) in Germany. ASR of incidence ranged from 26165.7 (23452.9–29365.4) in Fiji to 16.6 (14.5–19.1) in Germany. The top ten countries with highest age-standardised DALY rates were Fiji, Guam, Tonga, Tuvalu, Northern Mariana Islands, Palau, Solomon Islands, Papua New Guinea, Niue, Tokelau, respectively. The ASRs of prevalence and incidence were also high in these ten countries (Fig 2). China, with highest DALY count, ranked 24th in terms of ASR (124.6 [69.9–206.0]). Regarding the relationship between SDI and the ASRs of DALYs, prevalence, and incidence at the national level, it was observed that as SDI increased, these ASRs followed a pattern of initially rising and then declining, with the highest ASRs primarily concentrated in countries with medium to upper-middle SDI levels. This pattern was consistent with the trends previously described in the SDI regional level (S2 Fig).

Between 1990 and 2021, the largest EAPCs in ASR were observed in Sri Lanka (DALYs: 2.27 [2.26 to 2.28]; Prevalence: 2.27 [2.21 to 2.29]; Incidence: 2.13 [2.07 to 2.15]), the United States of America (DALYs: 1.28 [1.34 to 1.17]; Prevalence: 1.27 [1.55 to 1.01]; Incidence: 1.32 [1.53 to 1.11]), and Mexico (DALYs: 1.07 [1.08 to 1.00]; Prevalence: 1.06 [1.07 to 1.10]; Incidence: 1.06 [1.04 to 1.05]) (Fig 3).

**Table 2. Scabies prevalence in 2021 and trends from 1990 to 2021.**

| Characteristics | Prevalence | | | | |
|---|---|---|---|---|---|
| | Number (thousands) in 2021 (95% UI) | Percentage change (%) in number, 1990–2021 (95% UI) | ASR (per 100000) in 2021 (95% UI) | Percentage change (%) in ASR, 1990–2021 (95% UI) | EAPC of ASR (95% CI) |
| Global | 206549.6 (184175.5 to 231740.5) | 35.0 (31.1 to 39.0) | 2666.5 (2368.2 to 2994.5) | -2.6 (-3.4 to -1.6) | -0.02 (-0.02 to -0.03) |
| Sex | | | | | |
| Male | 105521.3 (93990.1 to 118412.6) | 33.9 (30.2 to 37.9) | 2701.2 (2397.0 to 3033.4) | -3.2 (-4.1 to -2.3) | -0.05 (-0.05 to -0.06) |
| Female | 101028.4 (90103.2 to 118412.6) | 36.1 (32.2 to 40.4) | 2631.5 (2341.6 to 2959.7) | -2.0 (-3.0 to -1.0) | -0.00 (0.00 to -0.02) |
| SDI | | | | | |
| High SDI | 5938.6 (5327.9 to 6604.6) | 32.9 (27.5 to 38.8) | 575.5 (513.5 to 647.2) | 9.0 (6.4 to 11.6) | 0.55 (0.59 to 0.51) |
| High-middle SDI | 33557.8 (29927.7 to 37366.6) | 21.9 (16.1 to 28.4) | 2748.9 (2443.6 to 3093.7) | 7.2 (5.7 to 8.9) | 0.18 (0.20 to 0.16) |
| Middle SDI | 89185.7 (79610.6 to 99815.6) | 24.6 (20.2 to 29.6) | 3774.7 (3361.3 to 4223.8) | -5.1 (-5.8 to -4.4) | -0.09 (-0.10 to -0.09) |
| Low-middle SDI | 55282.6 (48210.4 to 63002.9) | 45.1 (40.4 to 49.3) | 2820.5 (2488.6 to 3177.0) | -6.0 (-6.8 to -5.3) | -0.09 (-0.08 to -0.08) |
| Low SDI | 22395.2 (19434.9 to 25751.1) | 100.2 (96.3 to 103.8) | 1879.7 (1662.8 to 2122.7) | -8.1 (-9.1 to -7.1) | -0.27 (-0.30 to -0.25) |
| Region | | | | | |
| Andean Latin America | 2431.4 (2141.0 to 2760.9) | 58.3 (52.0 to 64.9) | 3712.9 (3266.9 to 4211.0) | 0.0 (-1.8 to 2.1) | -0.00 (0.00 to -0.01) |
| Australasia | 34.6 (30.8 to 38.9) | 42.6 (38.0 to 48.1) | 126.4 (110.0 to 144.1) | 0.6 (-1.8 to 3.2) | 0.30 (0.42 to 0.20) |
| Caribbean | 1705.1 (1513.4 to 1913.8) | 25.3 (21.4 to 29.8) | 3721.0 (3282.9 to 4208.2) | 0.2 (-1.1 to 1.4) | -0.00 (-0.01 to -0.01) |
| Central Asia | 546.7 (483.2 to 617.9) | 31.0 (27.3 to 35.2) | 576.0 (507.6 to 650.2) | 0.0 (-1.6 to 1.3) | -0.00 (-0.00 to 0.00) |
| Central Europe | 1574.6 (1413.9 to 1751.9) | -11.5 (-14.4 to -8.1) | 1469.0 (1307.3 to 1653.9) | 0.9 (-0.4 to 2.0) | 0.04 (0.03 to 0.03) |
| Central Latin America | 8166.8 (7202.8 to 9252.7) | 48.7 (42.7 to 55.2) | 3302.2 (2905.8 to 3733.9) | 9.0 (7.4 to 10.2) | 0.55 (0.55 to 0.56) |
| Central Sub-Saharan Africa | 1388.1 (1208.7 to 1584.1) | 149.7 (143.8 to 155.0) | 975.2 (865.4 to 1100.8) | 0.4 (-1.4 to 2.0) | 0.01 (-0.01 to 0.01) |
| East Asia | 65861.2 (58746.3 to 73336.4) | 9.8 (4.2 to 16.2) | 4806.8 (4279.4 to 5402.3) | 0.1 (-0.8 to 1.1) | -0.00 (-0.00 to -0.01) |
| Eastern Europe | 561.8 (501.9 to 626.7) | -12.6 (-15.1 to -10.2) | 307.6 (272.5 to 348.1) | 3.4 (2.5 to 4.3) | 0.11 (0.11 to 0.11) |
| Eastern Sub-Saharan Africa | 8337.8 (7175.9 to 9635.1) | 99.3 (92.0 to 105.9) | 1821.7 (1615.9 to 2057.4) | -9.4 (-11.9 to -7.0) | -0.67 (-0.76 to -0.58) |
| High-income Asia Pacific | 123.8 (111.0 to 137.6) | -4.5 (-9.0 to 0.9) | 78.2 (68.1 to 89.4) | 0.9 (-1.0 to 2.6) | 0.02 (0.03 to 0.02) |
| High-income North America | 449.9 (411.9 to 490.6) | 13.0 (7.5 to 19.8) | 132.7 (120.8 to 146.4) | -7.3 (-11.7 to -1.7) | 1.12 (1.38 to 0.89) |
| North Africa and Middle East | 6354.4 (5560.2 to 7215.1) | 64.8 (59.7 to 70.7) | 1009.7 (888.8 to 1139.2) | -3.6 (-6.1 to -1.3) | -0.14 (-0.24 to -0.06) |
| Oceania | 1146.0 (1008.8 to 1299.6) | 106.2 (100.5 to 111.5) | 7899.3 (7022.8 to 8880.3) | -0.8 (-2.7 to 1.2) | -0.05 (-0.05 to -0.05) |
| South Asia | 57565.3 (49643.2 to 66198.0) | 55.9 (50.6 to 60.5) | 3073.5 (2678.0 to 3502.6) | -0.1 (-0.9 to 0.5) | 0.23 (0.24 to 0.22) |
| Southeast Asia | 31819.4 (28357.2 to 35913.2) | 36.8 (31.9 to 42.2) | 4688.6 (4169.3 to 5280.7) | 0.1 (-0.6 to 0.8) | 0.06 (0.06 to 0.06) |

*(Continued)*

**Table 2.** (Continued)

| Characteristics | Prevalence | | | | |
| --- | --- | --- | --- | --- | --- |
| | Number (thousands) in 2021 (95% UI) | Percentage change (%) in number, 1990–2021 (95% UI) | ASR (per 100000) in 2021 (95% UI) | Percentage change (%) in ASR, 1990–2021 (95% UI) | EAPC of ASR (95% CI) |
| Southern Latin America | 49.3 (43.3 to 55.7) | 29.9 (25.3 to 34.1) | 76.1 (66.3 to 86.8) | 1.0 (-1.0 to 3.2) | 0.02 (0.03 to 0.01) |
| Southern Sub-Saharan Africa | 816.0 (718.7 to 928.6) | 46.5 (42.6 to 51.1) | 1001.4 (887.3 to 1132.0) | -0.1 (-1.0 to 0.9) | -0.00 (-0.00 to -0.01) |
| Tropical Latin America | 12817.0 (11464.8 to 14331.3) | 36.5 (30.5 to 43.3) | 5963.1 (5310.6 to 6767.3) | 0.1 (-0.7 to 0.9) | -0.12 (-0.11 to -0.11) |
| Western Europe | 53.5 (47.5 to 60.0) | 5.3 (2.1 to 8.4) | 14.3 (12.4 to 16.3) | -0.1 (-1.3 to 1.1) | 0.11 (0.12 to 0.10) |
| Western Sub-Saharan Africa | 4746.8 (4182.8 to 5448.4) | 153.5 (151.2 to 155.8) | 929.2 (829.0 to 1043.1) | -0.8 (-1.4 to -0.0) | -0.05 (-0.06 to -0.05) |

## Burden of scabies in 25 age groups in 2021

**Global level.** The numbers of DALYs, prevalence, and incidence peaked in the 5–9 age group, followed by a continuously decreasing trend (Fig 4A–4C). DALY and prevalence rates were higher from age 6 months to 24 years. However, incidence rate was higher in children aged 0–4 years, with the highest rates observed in the youngest age group. Additionally, the DALY, prevalence, and incidence rates gradually declined after adulthood, whereas serial increases among the elderly aged 60–89 years, particularly in incidence rate (Fig 4D–4F).

**SDI regional.** The distribution patterns of age-specific scabies burden varied across five SDI regions. In terms of quantity, the peaks of DALYs, prevalence, and incidence in low, low-middle and middle SDI regions occurred during 5–9 age group, similar to the pattern at global level. In high-middle and high SDI regions, the numbers of DALY, prevalence, and incidence were higher from ages 2 to 59 years, and lower in infants under 1 year and adults over 60 years (Fig 5A–5C).

In terms of age specific rates, the DALY, prevalence, and incidence rates were evenly distributed across all age groups in high SDI regions, with a relative peak occurring young people aged 20–24 years. In high-middle SDI regions, DALY and prevalence rates had two peaks, observed in the age groups of 12–23 months and 20–24 years. Incidence rate declined gradually from birth, increased moderately between ages 15 and 24, and then continued to decrease with minor fluctuations. In middle SDI regions, DALY and prevalence rates peaked in infants aged 12–23 month, followed by an overall decreasing trend, then showed a steady increase to a high level from the age of 60. Incidence rate showed a consistent decline from birth, with minor fluctuations, until age 60, after which a continuous increase was observed. In low-middle and low SDI regions, among relatively younger population, DALY and prevalence rates were higher between ages 6 months and 24 years, incidence rate was higher from birth to age 24. Among relatively older population, the rates (including DALYs, prevalence, and incidence) increased from age 60, reaching their highest levels in 95+ years age group. Notably, prevalence and incidence rates in the elderly over 95 years were highest across all age groups (Fig 5D–5F).

## Changes of scabies burden by age and sex from 1990 to 2021

**Global level.** The global scabies burden across different age groups has changed. Between 1990 and 2021, the numbers of DALYs, prevalence, and incidence declined before the age of two, after which they were increased. These increases became more pronounced with age. In the 95+ age group, DALY, prevalence, and incidence numbers increased by 617.5% (603.4–635.0), 618.1% (604.9–634.6), and 617.3% (603.1–634.6), respectively (Fig 6A–6C).

**Table 3. Scabies incidence in 2021 and trends from 1990 to 2021.**

| Characteristics | Incidence | | | | |
| --- | --- | --- | --- | --- | --- |
| | Number (thousands) in 2021 (95% UI) | Percentage change (%) in number, 1990–2021 (95% UI) | ASR (per 100000) in 2021 (95% UI) | Percentage change (%) in ASR, 1990–2021 (95% UI) | EAPC of ASR (95% CI) |
| Global | 622473.6 (556234.9 to 694992.0) | 34.5 (30.8 to 38.7) | 8049.5 (7165.2 to 9024.2) | -2.7 (-3.5 to -1.7) | -0.03 (-0.01 to -0.05) |
| Sex | | | | | |
| Male | 318054.3 (284005.7 to 355054.5) | 33.4 (29.7 to 37.4) | 8155.1 (7249.1 to 9139.2) | -3.3 (-4.2 to -2.4) | -0.05 (-0.04 to -0.07) |
| Female | 304419.2 (271886.0 to 339619.7) | 35.7 (31.7 to 40.2) | 7942.9 (7072.7 to 8918.2) | -2.1 (-3.1 to -1.1) | -0.01 (0.02 to -0.02) |
| SDI | | | | | |
| High SDI | 17877.2 (16076.5 to 19879.4) | 32.5 (27.5 to 38.3) | 1737.0 (1552.8 to 1938.0) | 8.6 (6.0 to 11.3) | 0.54 (0.56 to 0.49) |
| High-middle SDI | 101065.7 (90201.5 to 112619.0) | 21.5 (15.7 to 27.9) | 8312.8 (7413.4 to 9311.0) | 7.1 (5.6 to 8.7) | 0.18 (0.20 to 0.16) |
| Middle SDI | 268680.3 (240226.2 to 299946.2) | 24.1 (19.7 to 29.1) | 11409.1 (10174.5 to 12796.2) | -5.2 (-5.9 to -4.4) | -0.09 (-0.10 to -0.10) |
| Low-middle SDI | 166777.8 (146956.6 to 189229.2) | 44.4 (40.1 to 48.4) | 8521.7 (7528.2 to 9581.9) | -6.0 (-6.8 to -5.3) | -0.08 (-0.06 to -0.08) |
| Low SDI | 67501.1 (58688.5 to 77417.4) | 100.0 (96.2 to 103.7) | 5648.4 (5034.5 to 6348.3) | -7.8 (-8.9 to -6.9) | -0.26 (-0.26 to -0.25) |
| Region | | | | | |
| Andean Latin America | 7314.4 (6474.1 to 8227.9) | 57.8 (51.1 to 64.7) | 11185.7 (9924.1 to 12525.4) | 0.0 (-1.9 to 2.1) | -0.00 (-0.00 to -0.00) |
| Australasia | 103.7 (91.9 to 115.3) | 42.6 (38.2 to 47.8) | 379.6 (333.0 to 430.5) | 0.6 (-1.9 to 3.3) | 0.27 (0.35 to 0.18) |
| Caribbean | 5128.0 (4564.3 to 5739.4) | 25.1 (21.3 to 29.2) | 11210.7 (9923.8 to 12581.0) | 0.2 (-1.3 to 1.4) | -0.00 (0.01 to 0.01) |
| Central Asia | 1654.1 (1469.5 to 1870.4) | 30.6 (27.0 to 34.7) | 1744.7 (1547.5 to 1976.1) | -0.0 (-1.5 to 1.4) | -0.00 (-0.01 to -0.01) |
| Central Europe | 4722.5 (4224.6 to 5271.9) | -11.6 (-14.4 to -8.3) | 4418.6 (3943.3 to 4984.5) | 0.9 (-0.3 to 2.0) | 0.04 (0.04 to 0.04) |
| Central Latin America | 24584.6 (21794.0 to 27639.5) | 47.8 (42.0 to 53.8) | 9971.3 (8831.7 to 11244.9) | 8.9 (7.4 to 10.0) | 0.54 (0.53 to 0.54) |
| Central Sub-Saharan Africa | 4223.1 (3700.5 to 4819.4) | 148.4 (142.2 to 153.9) | 2954.4 (2646.5 to 3321.1) | 0.4 (-1.4 to 2.1) | 0.00 (-0.00 to 0.01) |
| East Asia | 198512.8 (177211.1 to 220994.2) | 9.4 (3.6 to 15.3) | 14562.3 (12977.0 to 16303.0) | 0.1 (-0.9 to 1.1) | -0.00 (-0.01 to 0.00) |
| Eastern Europe | 1680.6 (1508.3 to 1866.9) | -12.6 (-15.1 to -10.1) | 923.7 (820.2 to 1045.3) | 3.4 (2.6 to 4.3) | 0.11 (0.11 to 0.12) |
| Eastern Sub-Saharan Africa | 24949.3 (21737.8 to 28779.6) | 100.8 (94.2 to 106.9) | 5431.6 (4846.5 to 6106.4) | -8.4 (-10.7 to -6.0) | -0.63 (-0.69 to -0.58) |
| High-income Asia Pacific | 372.9 (333.3 to 413.9) | -4.4 (-9.1 to 0.8) | 236.7 (208.0 to 270.2) | 0.9 (-0.8 to 2.4) | 0.02 (0.02 to 0.02) |
| High-income North America | 1360.5 (1237.6 to 1481.6) | 12.3 (7.2 to 18.2) | 402.4 (364.6 to 445.8) | -7.7 (-12.0 to -2.6) | 1.16 (1.35 to 0.96) |
| North Africa and Middle East | 19137.8 (16887.3 to 21828.8) | 64.5 (59.3 to 70.4) | 3044.7 (2694.4 to 3454.7) | -3.3 (-5.4 to -1.2) | -0.13 (-0.20 to -0.06) |
| Oceania | 3460.6 (3073.7 to 3928.8) | 106.1 (100.2 to 111.2) | 23783.1 (21294.9 to 26555.7) | -0.7 (-2.7 to 1.3) | -0.04 (-0.05 to -0.07) |
| South Asia | 173638.1 (151183.2 to 199068.5) | 54.8 (49.7 to 59.3) | 9304.1 (8141.0 to 10583.3) | -0.1 (-0.8 to 0.6) | 0.23 (0.25 to 0.22) |
| Southeast Asia | 96003.1 (85497.3 to 108062.8) | 36.4 (31.4 to 41.5) | 14179.0 (12604.8 to 16064.5) | 0.1 (-0.7 to 0.9) | 0.06 (0.05 to 0.07) |

(*Continued*)

**Table 3.** (Continued)

| Characteristics | Incidence | | | | |
|---|---|---|---|---|---|
| | Number (thousands) in 2021 (95% UI) | Percentage change (%) in number, 1990–2021 (95% UI) | ASR (per 100000) in 2021 (95% UI) | Percentage change (%) in ASR, 1990–2021 (95% UI) | EAPC of ASR (95% CI) |
| Southern Latin America | 148.3 (130.9 to 168.2) | 29.2 (24.4 to 33.7) | 229.9 (201.0 to 262.3) | 1.0 (-1.2 to 3.3) | 0.02 (0.01 to -0.00) |
| Southern Sub-Saharan Africa | 2468.6 (2191.2 to 2811.3) | 45.8 (41.6 to 50.5) | 3033.2 (2703.1 to 3437.1) | -0.1 (-1.0 to 0.9) | -0.00 (-0.00 to -0.00) |
| Tropical Latin America | 38519.3 (34603.3 to 43240.2) | 37.0 (30.9 to 43.5) | 17921.1 (16005.4 to 20289.7) | 0.1 (-0.7 to 1.0) | -0.11 (-0.11 to -0.11) |
| Western Europe | 158.9 (142.5 to 177.5) | 5.5 (2.5 to 8.4) | 42.6 (37.3 to 48.9) | 0.1 (-1.3 to 1.3) | 0.11 (0.12 to 0.13) |
| Western Sub-Saharan Africa | 14332.4 (12626.1 to 16444.5) | 152.5 (150.0 to 154.9) | 2792.7 (2510.1 to 3116.9) | -0.8 (-1.4 to -0.0) | -0.05 (-0.05 to -0.05) |

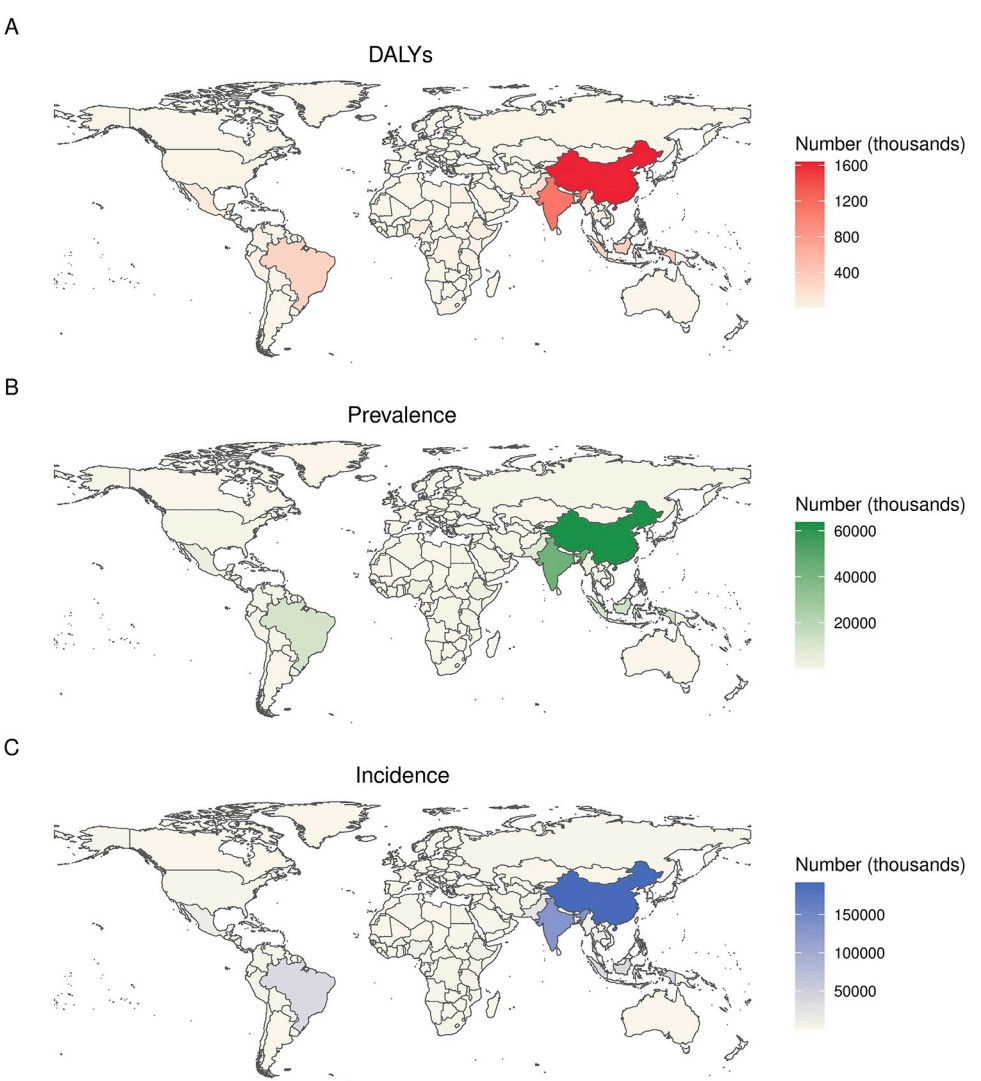

**Fig 1.   World map showing the numbers of scabies DALYs (A), prevalence (B), and incidence (C) in 2021.**
Basemap data source: Natural Earth, http://www.naturalearthdata.com, public domain.

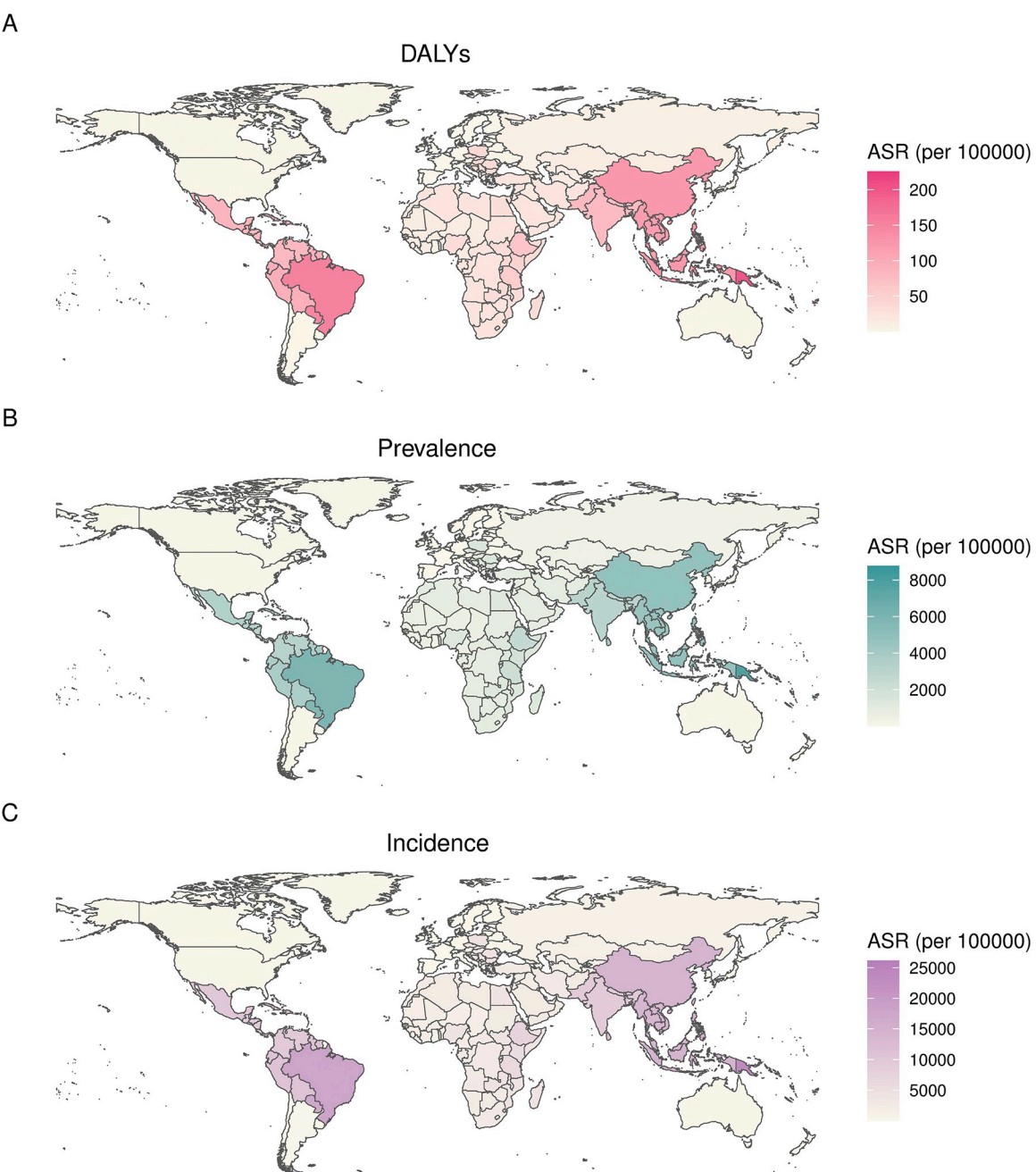

**Fig 2. World map showing ASRs of scabies DALYs (A), prevalence (B), and incidence (C) in 2021.** Basemap data source: Natural Earth, http://www.naturalearthdata.com, public domain.

The DALYs, prevalence, and incidence rates generally decreased before the age of 40, except for a slight increase at ages 30–34. Since the age of 40, these rates have increased, especially for women. After the age of 75, the increases became more substantial. In the age group of 90–94 years, DALY, prevalence, and incidence rates increased by 38.1% (35.7–40.7), 38.2% (36.2–40.4), and 38.1% (36.3–40.3), respectively. For females, these rates increased by 49.8% (46.8–53.4), 50.4% (47.6–53.7), and 50.3% (47.8–53.5). For males, these rates increased by 16.4% (13.9–19.1), 15.8% (13.8–18.1), and 15.7% (13.8–17.9) (Fig 6D–6F).

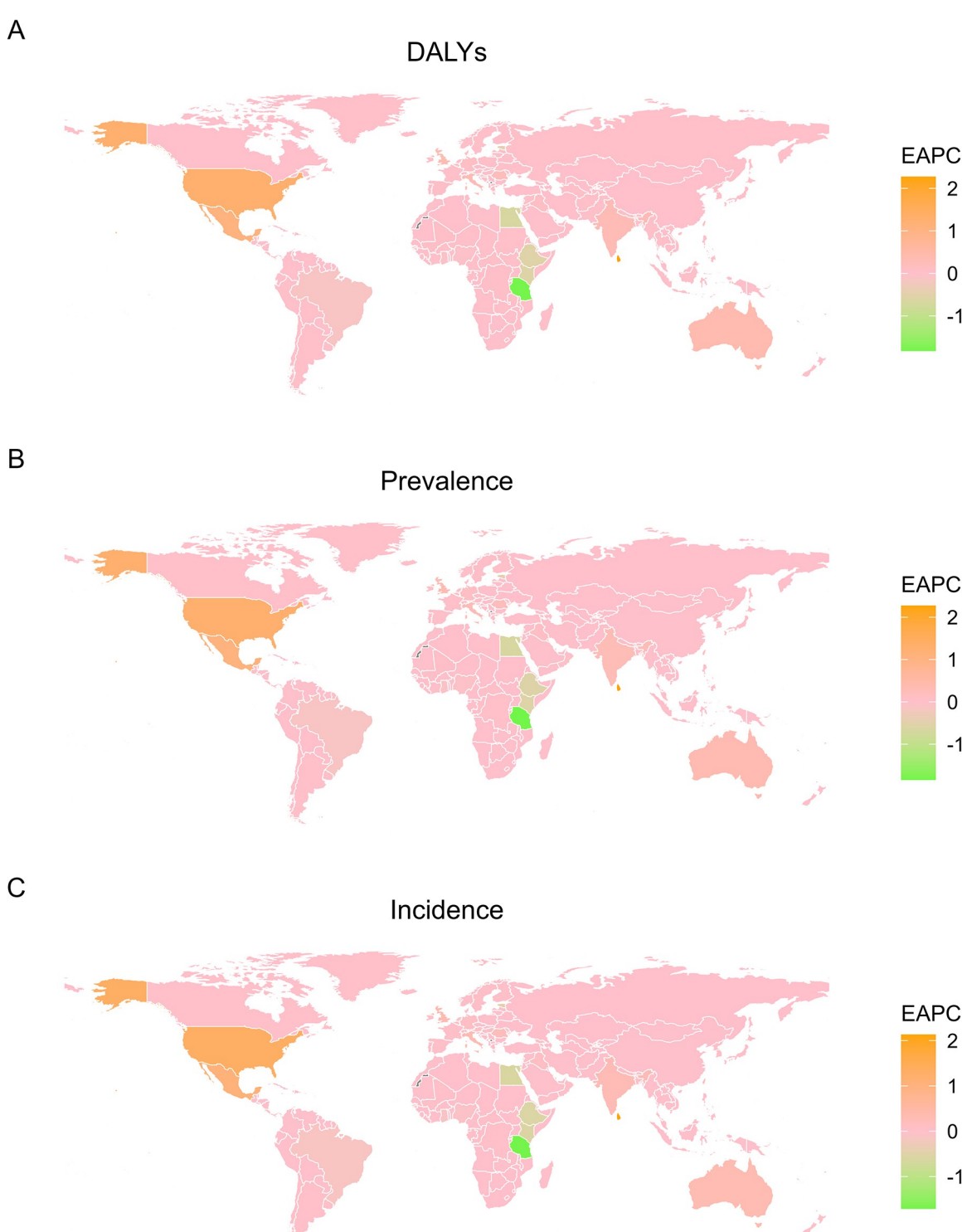

**Fig 3. World map showing EAPCs in scabies DALY (A), prevalence (B), and incidence ASRs (C) from 1990 to 2021.** Basemap data source: Natural Earth, http://www.naturalearthdata.com, public domain.

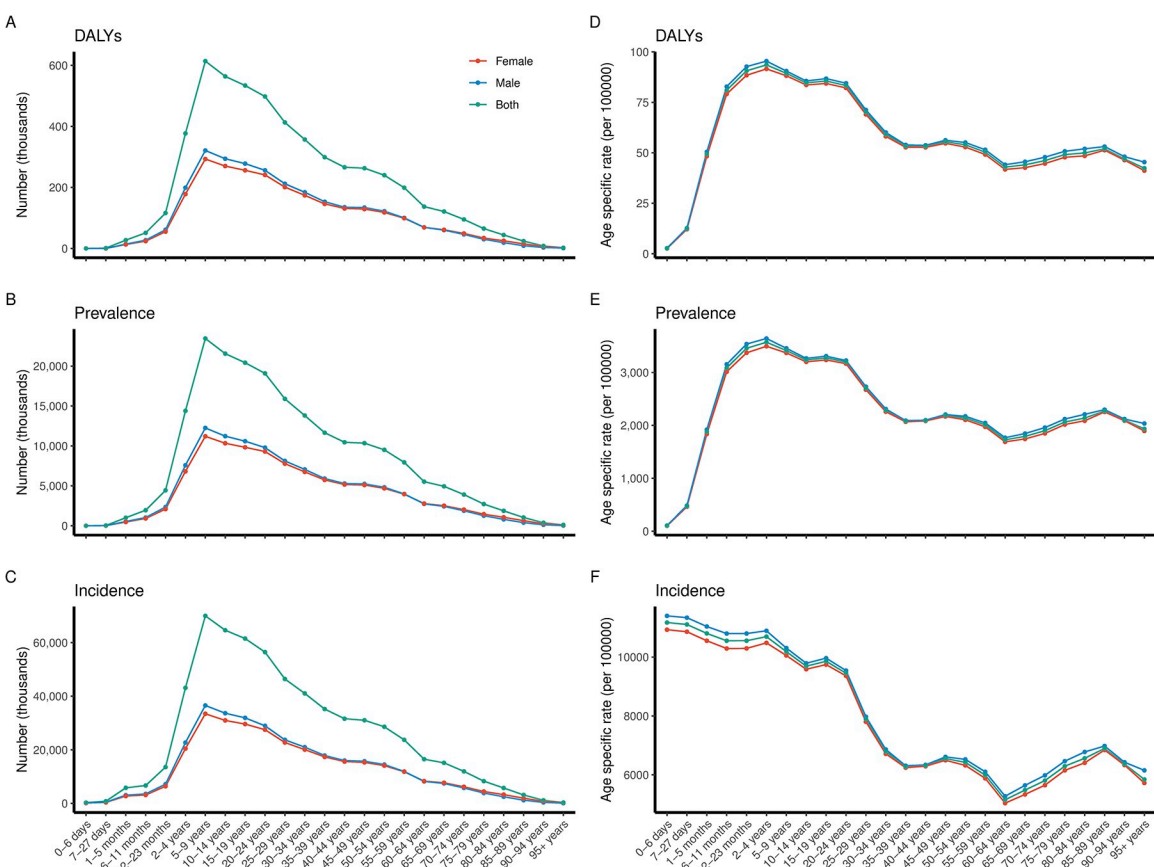

**Fig 4.** Global scabies DALYs (A, D), prevalence (B, E), and incidence (C, F) by age and sex in 2021.

**SDI regional.** In high SDI regions, DALY, prevalence, and incidence rates showed an upward trend from age 20. Compared to males, a greater increase was observed among females in the elderly aged 60–95 years. In high-middle SDI regions, these rates increased, excluding the ages less than 1 years and 15–24 years, with a generally greater rise observed among females than males. In middle SDI regions, DALY, prevalence, and incidence rates increased after the age of 70. In low-middle SDI regions, there was a relatively obvious increase in these rates among the elderly after 70 years. In low SDI regions, DALY, prevalence, and incidence rates rose after the age of 65 (S3–S7 Figs).

## Discussion

Our study systematically examined the burden and epidemiological characteristics of scabies from three dimensions: geographical location, gender, and age. Although the age-standardised DALY, prevalence, and incidence rates of scabies have declined from 1990 to 2021, the absolute number continued to rise, indicating that the burden in 2021 remained at a concerning level. In 2021, among the five SDI regions, the ASRs of DALY, prevalence, and incidence were highest in middle SDI regions and lowest in high SDI regions. Among the 21 geographical regions, the heaviest burden was observed in Oceania, tropical Latin America, and East Asia. At national level, Fiji ranked first in age-standardised DALY, prevalence, and incidence rates caused by scabies. Across all age groups, children and young people bore a heavier scabies burden, and a slight resurgence was observed from age 60 to 89. Between 1990 and 2021, the

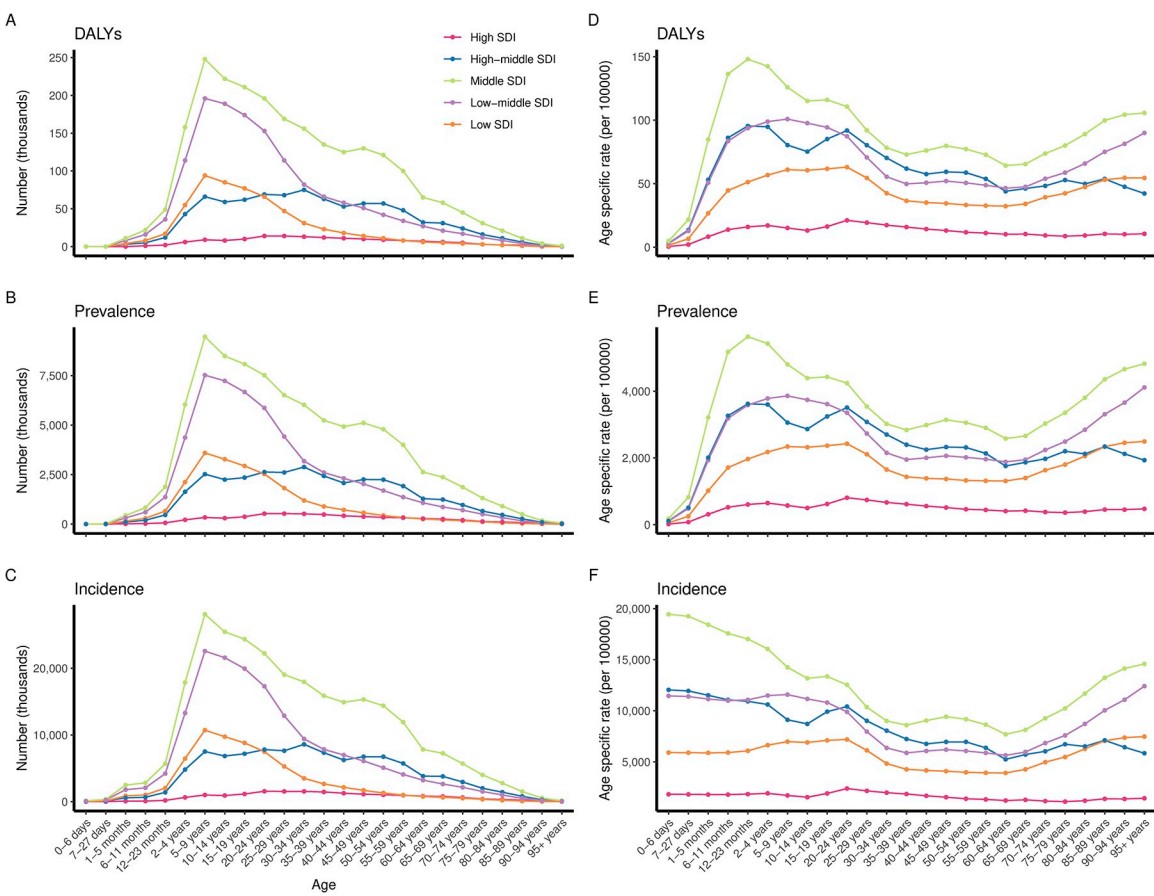

**Fig 5.** Regional scabies DALYs (A, D), prevalence (B, E), and incidence (C, F) by age in 2021.

burden increased in both high and high-middle SDI regions. A relatively obvious increase was observed in Central Latin America and high-income North America compared to other regions. The DALY, prevalence, and incidence rates among adults aged 40 and above have risen, with a more pronounced increase in the elderly and females.

The scabies burden was lowest in high SDI regions, possibly due to the overall higher standards in economics, society, and environment in these areas. Previous literature has indicated a close association between scabies occurrence and factors such as crowded impoverished living conditions, contaminated water sources, and limited access to healthcare [14], advantages which high SDI regions tended to possess. Over the period from 1990 to 2021, while the burden of scabies declined in low, low-middle, and middle SDI regions, it increased in high and high-middle SDI regions. In the past five decades, the global average SDI has risen. Lower SDI regions advanced more rapidly [15], which may promote health enhancements and reduction of burden in these regions. Signs of an increasing burden in higher SDI regions have emerged, with some recent studies [5,16,17] noting its growing prevalence in developed high-income countries. This rise may be attributed to factors such as population mobility, aging demographics, and an increase in populations affected by sexually transmitted diseases [16]. In older populations, obesity may exacerbate infections, while dementia or physical limitations caused by chronic-degenerative diseases may delay early diagnosis [18]. This issue warrants further investigation in future research. According to reports, scabies cases have been steadily rising in Germany since 2009. As scabies was not reportable in Germany, estimates of cases

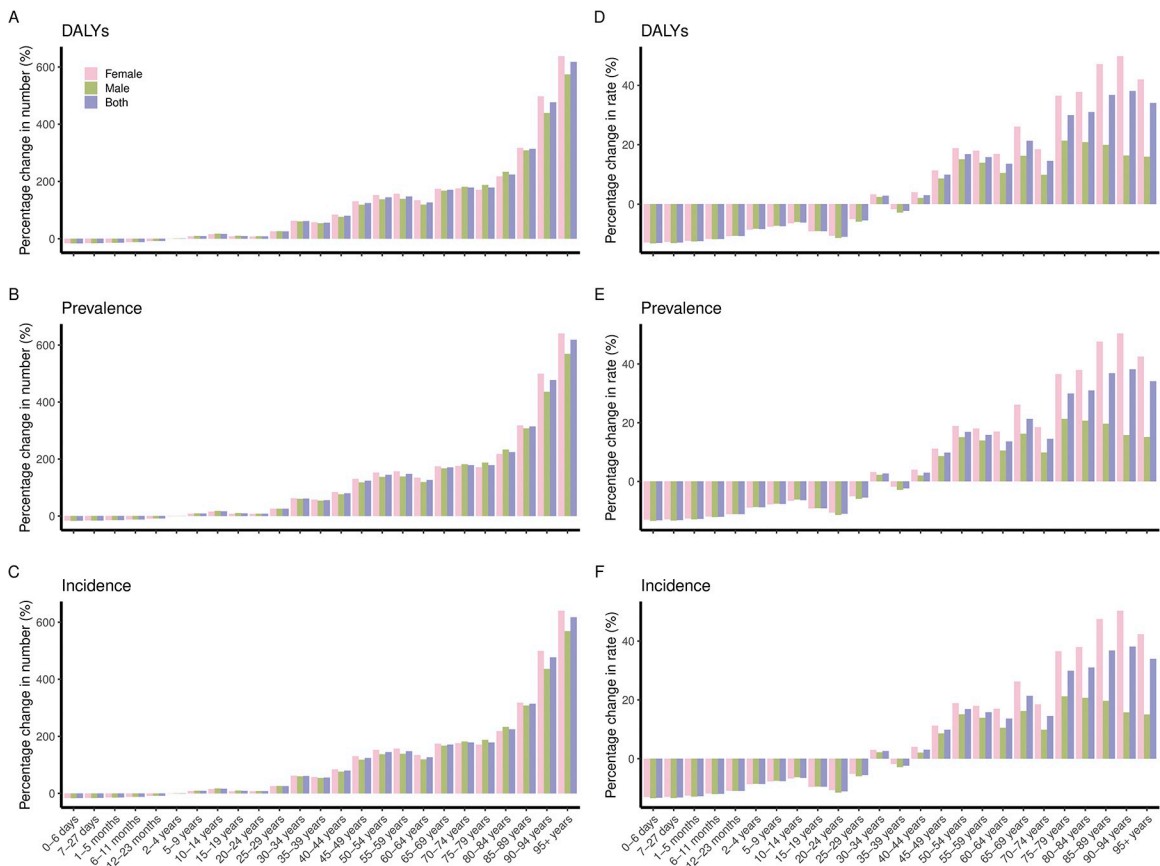

**Fig 6.** Percentage changes in scabies DALYs (A, D), prevalence (B, E), and incidence (C, F) by age and sex from 1990 to 2021.

were derived from self-reported scabies, records of drug purchases, statistics from health insurance companies, etc., suggesting that actual figures may be higher [3,11,17].

In 2021, the burden of scabies in Oceania, tropical Latin America, East Asia, and Southeast Asia were higher, consistent with previous findings [7]. At national level, the high burden in Fiji, Papua New Guinea, and Solomon Islands has been widely pointed. In Fiji, mass drug administration has been actively implemented to control scabies [19,20]. From 1990 to 2021, DALY burden in Central Latin America, high-income North America, Eastern Europe, and Central Europe increased. The rising burden in Central Latin America may be attributed to its geographical environment and challenging socioeconomic conditions [21]. In addition to the risk factors in higher SDI regions mentioned above, research suggested that the increasing prevalence in some European countries may be linked to treatment failures. Predictors of treatment failure include the immune status of the host, the choice of treatment, exposure to future transmission events, and potential drug resistance. Although no conclusive evidence currently exists to confirm scabies resistance to frontline scabicides, such as permethrin and ivermectin, skepticism about their efficacy has grown in recent years, highlighting the need for further studies to investigate the underlying causes of treatment failure [5]. Furthermore, travel and refugee movements contribute to the spread of scabies in European regions, driven by population influxes from areas with high scabies prevalence and overcrowded living conditions that promote direct and prolonged skin-to-skin contact [3].

In 2021, the primary drivers of global scabies burden remained children and young people. However, the burden among this demographic has declined from 1990 to 2021. Unexpectedly, there was an increase in burden among the population aged over 40, with a particularly significant increase observed in the elderly. In high and high-middle SDI regions, scabies burden shown a growth since adulthood. In low, low-middle, and middle SDI regions, only a smaller increase in burden was observed among the elderly. As the aging population accelerates, the proportion of adults over 40 gradually rises [22], potentially leading to increased scabies incidence in this age group. Especially in high-income developed countries, such as Japan, the population aged 65 and above surged by over 480% between 1950 and 2019 [15]. The weaker immunity in the elderly and proliferation of nursing homes may have further amplified the risk of scabies outbreaks [23–25].

Few studies have analysed trends for age-specific scabies occurrence in males and females over time. In our study, within age brackets witnessing an increase in scabies burden, females exhibited a faster growth compared to males. This may be attributed to the physiological differences between men and women, as well as shifts in women's lifestyles and social roles. Female skin tends to be thinner compared to males, particularly post-menopause, which might influence skin resistance [26]. However, further research is needed to confirm these observations and elucidate the underlying mechanisms. Additionally, increased work pressures on modern women and more frequent use of body care products may compromise skin barrier function [27]. Furthermore, women are more frequently employed in caregiving-related settings, such as various long-term care facilities, nursing homes, community units [28], and are more likely to reside in care facilities in old age [29], thereby increasing exposure risks to scabies.

This analysis still has some limitations. The quality and completeness of scabies data collected by GBD 2021 from various sources need further improvement [11]. It remains challenging to collect accurate scabies data in low SDI regions and due to migration [15]. Some countries, such as Germany and Japan [17,30], do not report scabies data. In resource-limited settings, many affected individuals do not seek treatment due to financial constraints and perceptions of care inefficacy. Consequently, reliance on routine clinical data often results in an underestimation of the disease burden. It is likely that the true global prevalence of scabies is considerably higher than current estimates [31]. Strengthening public awareness and providing better resources for prevention and treatment may help address these challenges in the future. When estimating scabies-related data, covariates are limited to SEV for unsafe water, SDI, sugar consumption, and the Healthcare Access and Quality index, which do not encompass all determinants of scabies prevalence [11,31]. The GBD methodology primarily focuses on the direct effects of scabies, such as itching and skin disfigurement, while failing to account for systemic sequelae resulting from secondary infections caused by Group A Streptococcus, including impetigo, glomerulonephritis, and rheumatic fever, which are frequently associated with scabies [31]. Furthermore, it does not capture mortality associated with crusted scabies [32–34], thus potentially underestimating the overall disease burden. More detailed limitation discussed in elsewhere [11,12]. Furthermore, this study lacks an analysis of scabies risk factors. Future directions may focus on analyzing scabies risk factors in different geographic regions, age structures, and genders to provide clearer guidance for global scabies control efforts.

Overall, the distribution of scabies burden in 2021 shared some similarities with previous studies, such as heavier burdens in tropical regions and among children and young people. However, over the past 30 years, the epidemiological characteristics of scabies have been changing. Several key points warrant particular attention: 1. The growth in scabies burden in high and high-middle SDI regions; 2. The rising burden among adults aged 40 and above, particularly the elderly; 3. The significant increase in scabies burden among women. In the future,

in addition to continuing to monitor the high burden of scabies due to socioeconomic under-development and hot climates, several factors should also be taken into consideration. These include substantial population movements such as migration and refugee activities, crowded and confined environments like orphanages, nursing homes, and long-term care facilities, as well as global demographic shifts like population aging, and decreasing treatment efficacy. Strategies such as large-scale oral administration of ivermectin, combination therapy with ivermectin and permethrin, ongoing clinical trials of moxidectin for scabies treatment, and the development of scabies vaccines may prove helpful in controlling scabies [5,16,17,21].

## Supporting information

**S1 Fig.** Scabies DALYs (A, D), prevalence (B, E), and incidence (C, F) each year from 1990 to 2021.
(TIF)

**S2 Fig.** Scabies DALYs (A), prevalence (B), and incidence (C) in ASR across 204 countries and territories by SDI in both sexes, 2021.
(TIF)

**S3 Fig.** Percentage changes in scabies DALYs (A, D), prevalence (B, E), and incidence (C, F) in high SDI regions from 1990 to 2021.
(TIF)

**S4 Fig.** Percentage changes in scabies DALYs (A, D), prevalence (B, E), and incidence (C, F) in high-middle SDI regions from 1990 to 2021.
(TIF)

**S5 Fig.** Percentage changes in scabies DALYs (A, D), prevalence (B, E), and incidence (C, F) in middle SDI regions from 1990 to 2021.
(TIF)

**S6 Fig.** Percentage changes in scabies DALYs (A, D), prevalence (B, E), and incidence (C, F) in low-middle SDI regions from 1990 to 2021.
(TIF)

**S7 Fig.** Percentage changes in scabies DALYs (A, D), prevalence (B, E), and incidence (C, F) in low SDI regions from 1990 to 2021.
(TIF)

## Acknowledgments

We extend our sincere gratitude to the GBD 2021 study collaborators for their dedicated efforts in collecting scabies-related data.

## Author Contributions

**Conceptualization:** Zehu Liu, Xiujiao Xia.

**Data curation:** Jiajia Li.

**Formal analysis:** Jiajia Li.

**Funding acquisition:** Zehu Liu.

**Project administration:** Zehu Liu.

**Visualization:** Jiajia Li.

**Writing – original draft:** Jiajia Li.

**Writing – review & editing:** Zehu Liu, Xiujiao Xia.

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
