## [Decision Letter · Decision Letter 0]

23 Sep 2024

Dear doctor Li,

Thank you very much for submitting your manuscript "The disability-adjusted life years (DALYs), prevalence and incidence of scabies, 1990–2021: a systematic analysis from the Global Burden of Disease Study 2021" for consideration at PLOS Neglected Tropical Diseases. As with all papers reviewed by the journal, your manuscript was reviewed by members of the editorial board and by several independent reviewers. In light of the reviews (below this email), we would like to invite the resubmission of a significantly-revised version that takes into account the reviewers' comments. 

We cannot make any decision about publication until we have seen the revised manuscript and your response to the reviewers' comments. Your revised manuscript is also likely to be sent to reviewers for further evaluation.

Sincerely,

Benedikt Ley, PhD

Guest Editor

Jong-Yil Chai

Section Editor

Reviewer's Responses to Questions

**Key Review Criteria Required for Acceptance?**

**Methods**

-Are the objectives of the study clearly articulated with a clear testable hypothesis stated?

-Is the study design appropriate to address the stated objectives?

-Is the population clearly described and appropriate for the hypothesis being tested?

-Is the sample size sufficient to ensure adequate power to address the hypothesis being tested?

-Were correct statistical analysis used to support conclusions?

-Are there concerns about ethical or regulatory requirements being met?

Reviewer #1: The study reported the disease burden only at the beginning and end of the time period without addressing the trends in between. Please include an analysis of the trends over time using the Estimated Annual Percentage Change (EAPC).

Instead of a simple hierarchical analysis, please add the correlation between disease burden and Sociodemographic Index (SDI) at the national level.

Reviewer #2: (No Response)

Reviewer #3: • It would be good to clearly articulate what variables from the GBD databased were used for this analysis

• The methods section seems to lack an explanation how DALYs were calculated.

•

**Results**

-Does the analysis presented match the analysis plan?

-Are the results clearly and completely presented?

-Are the figures (Tables, Images) of sufficient quality for clarity?

Reviewer #1: (No Response)

Reviewer #2: (No Response)

Reviewer #3: • I find it a bit confusing to talk about ‘prevalent cases’ rather than just ‘prevalence’ (same for incidence) and suggest to change wording unless the authors feel strongly about it. Similarly referring to a ‘low burden’ as ‘light burden’ seems odd. Consider changing.

• I am wondering if it would be possible to add a temporal analysis based on existing control strategies in the different countries.

**Conclusions**

-Are the conclusions supported by the data presented?

-Are the limitations of analysis clearly described?

-Do the authors discuss how these data can be helpful to advance our understanding of the topic under study?

-Is public health relevance addressed?

Reviewer #1: (No Response)

Reviewer #2: (No Response)

Reviewer #3: (No Response)

**Editorial and Data Presentation Modifications?**

Reviewer #1: (No Response)

Reviewer #2: (No Response)

Reviewer #3: (No Response)

**Summary and General Comments**

Reviewer #1: (No Response)

Reviewer #2: I appreciate the opportunity to critically review the manuscript titled “The disability-adjusted life years (DALYs), prevalence and incidence of scabies, 1990–2021: a systematic analysis from the Global Burden of Disease Study 2021” in which, as the title indicates, the researchers analyzed the most recent GBD data related to scabies. The manuscript is well-written and adheres closely to the methodology of a systematic analysis of GBD data. I present the authors with a series of comments, all of which are minor.

Comment 1: I agree that the GBD methodology, particularly for 2021, has been extensively described in previous publications. However, it is always worthwhile to include a couple of general lines to briefly describe the estimation of the metrics of interest.

Comment 2: What is the disability weight for scabies? If it is low, what are the implications for the obtained metrics? I suggest briefly citing this information.

Comment 3: Lines 352-353 briefly discuss the factors that may be driving the increase in scabies burden in high and upper-middle SDI countries. This is a very important finding that I believe warrants further development. For example, what role do obesity and chronic-degenerative diseases play? What about opportunities for earlier diagnosis in these regions?

Reviewer #3: Can the authors make clearer what the added benefit of their analysis to the already available results in the GBD results tool is?

 In the limitation section the authors state that the countries like Germany don’t report scabies data, but in the results section data for Germany is presented. Please clarify.

PLOS authors have the option to publish the peer review history of their article (what does this mean?). If published, this will include your full peer review and any attached files.

Reviewer #1: No

Reviewer #2: Yes: Efrén Murillo-Zamora

Reviewer #3: No
---

## [Decision Letter · Decision Letter 1]

28 Nov 2024

PNTD-D-24-00963R1The disability-adjusted life years (DALYs), prevalence and incidence of scabies, 1990–2021: a systematic analysis from the Global Burden of Disease Study 2021PLOS Neglected Tropical Diseases Dear Dr. Li, Thank you for submitting your manuscript to PLOS Neglected Tropical Diseases. After careful consideration, we feel that it has merit but does not fully meet PLOS Neglected Tropical Diseases's publication criteria as it currently stands. Therefore, we invite you to submit a revised version of the manuscript that addresses the points raised during the review process. Please submit your revised manuscript within 30 days Dec 28 2024 11:59PM. If you will need more time than this to complete your revisions, please reply to this message or contact the journal office at plosntds@plos.org. Please include the following items when submitting your revised manuscript:*
A rebuttal letter that responds to each point raised by the editor and reviewer(s). You should upload this letter as a separate file labeled 'Response to Reviewers'. This file does not need to include responses to any formatting updates and technical items listed in the 'Journal Requirements' section below.*
A marked-up copy of your manuscript that highlights changes made to the original version. You should upload this as a separate file labeled 'Revised Manuscript with Track Changes'.*
An unmarked version of your revised paper without tracked changes. You should upload this as a separate file labeled 'Manuscript'. If you would like to make changes to your financial disclosure, competing interests statement, or data availability statement, please make these updates within the submission form at the time of resubmission. Guidelines for resubmitting your figure files are available below the reviewer comments at the end of this letter. We look forward to receiving your revised manuscript. Kind regards, Jong-Yil ChaiSection EditorPLOS Neglected Tropical Diseases Jong-Yil ChaiSection EditorPLOS Neglected Tropical Diseases

Shaden Kamhawi

co-Editor-in-Chief

Paul Brindley

co-Editor-in-Chief

 **Additional Editor Comments :** Thanks for your revision of the manuscript. A few minor revisions are needed to be finally accepted for publication (see the comments of the review 6).  **Journal Requirements:** 1) Thank you for uploading your study's underlying data set. Unfortunately, the repository you have noted in your Data Availability statement does not qualify as an acceptable data repository according to PLOS's standards. At this time, please upload the minimal data set necessary to replicate your study's findings to a stable, public repository (such as figshare or Dryad) and provide us with the relevant URLs, DOIs, or accession numbers that may be used to access these data. For a list of recommended repositories and additional information on PLOS standards for data deposition, please see https://journals.plos.org/plosntds/s/recommended-repositories

 **Reviewers' comments:** Reviewer's Responses to Questions

**Key Review Criteria Required for Acceptance?**

**Methods**

-Are the objectives of the study clearly articulated with a clear testable hypothesis stated?

-Is the study design appropriate to address the stated objectives?

-Is the population clearly described and appropriate for the hypothesis being tested?

-Is the sample size sufficient to ensure adequate power to address the hypothesis being tested?

-Were correct statistical analysis used to support conclusions?

-Are there concerns about ethical or regulatory requirements being met?

Reviewer #2: Yes

Reviewer #4: Yes, the objective is clear and the study design is appropriate. The database has proven to be reliable and the statistical analysis is correct for supporting conclusions. I do not have any ethical or regulatory requirement for this manuscript.

Reviewer #5: The methodology is adequately described by which the Global Diseases Burden 2021 data were used, and the authors methodology is similar to that used by other published studies using older data from the GBD study (Zhang et al 2020 and Karimkhani et al 2017).

Reviewer #6: Introduction paragraph: The first paragraph gives a good description of scabies, though the paragraph is somewhat disjointed and I would encourage the authors to reorganize the information here. For instance, lines 85-86 seem out of place (though important to state).

Line 80: I don’t feel the authors need to include species specific bacteria here.

Line 84: is ‘resource-poor’ the preferred term?

Line 95-96: be more specific here; why do the authors feel the efficacy of the various treatment options warrants further investigation.

Methods and statistical analyses appear sound, though it may be advantageous to have a reviewer with more expertise in statistics.

Reviewer #7: The objectives of the study are clearly articulated, and the study design does address the stated objectives. Populations, sample size and other criteria are clearly defined, and statistical analysis appears to be accurate and appropriate to the study objectives.

**Results**

-Does the analysis presented match the analysis plan?

-Are the results clearly and completely presented?

-Are the figures (Tables, Images) of sufficient quality for clarity?

Reviewer #2: Yes

Reviewer #4: Yes. The results follow the analysis plan and the ones that were missing between years were added in the revision to provide a complete view. The figures are blurred in the PDF, but when they are downloaded they have sufficient quality for clarity. It is just a matter of the presentation.

Reviewer #5: The data are clearly presented and match the analysis plan. The figures are very helpful, good quality, and clearly labelled.

Reviewer #6: Results section: The authors present a detailed report of scabies DALYs over the past 3 decades using GBD data. Given the change in population over this period of time, it may be helpful to present incidence as cases per 100,000 people in the regional, national sections. Overall it is a detailed description of the global burden of scabies and the figures provide good representation of the data.

Line 200: I would suggest using another term than “most severe”

Reviewer #7: The results are clear, and match the analysis plan. All data provided is based and of high quality.

**Conclusions**

-Are the conclusions supported by the data presented?

-Are the limitations of analysis clearly described?

-Do the authors discuss how these data can be helpful to advance our understanding of the topic under study?

-Is public health relevance addressed?

Reviewer #2: Yes

Reviewer #4: Yes, the conclusions are totally supported by the data presented. The main problem for the scabies data analysis is the collection of data in low-income countries and the authors address the problem in the manuscript. They do analyze the results and discuss about how their study can be used to improve health outcomes in different countries.

Reviewer #5: The conclusions are measured and align with the data presented.

Reviewer #6: Lines 373-375: Authors state that Central America has had a higher burden of scabies and suggest it may be related to a decline in socioeconomic conditions. Though the paragraph starts with a discussion of scabies incidence on a national level. Can the authors report more on the country specific data in Central America and if this correlates with changes in socioeconomic status, recognizing that not all countries in Central America are suffering from suboptimal socioeconomic conditions.

Lines 378-382: authors make statements regarding the possibility of a decline in the efficacy of first line scabies treatments; this should be developed further in the discussion. In addition, I would suggest expanding the discussion on how scabies incidence may be impacted by migration.

403-404: authors suggest that post-menopausal women may have an increased risk of scabies due to changes in the skin; if the authors are going to include this here they need to provide more substantial evidence to support this claim.

411-425: Authors discuss the limitations of this study, and I would suggest further discussion of the impact of under-reporting of scabies, particularly in LMICs.

Reviewer #7: The conclusions are indeed supported by the presented data, and the limitations are clearly describes. The discussion on the data by the authors is comprehensive and provides essential point of view to understanding the topic. Public health relevance is properly addressed.

**Editorial and Data Presentation Modifications?**

Reviewer #2: (No Response)

Reviewer #4: Page 5, line 61. The sentence ends with "et al", but it does not make sense. I think that these words were misplaced and that they can be deleted to end the sentence with "unclear water sources".

Reviewer #5: The study is clearly written, and the authors appear to have addressed the concerns previously raised by my reviewing colleagues, including the additional analyses suggested by reviewer 1 including trends over time by SDI.

Reviewer #6: minor revision

Reviewer #7: No major recommendations from my end, only would note that et al. appears twice in the paper with no names preceding - lines 61 and 410.

**Summary and General Comments**

Reviewer #2: I extend my gratitude to the authors for considering my observations and suggestions. I congratulate them on their research.

Reviewer #4: The authors complied with the modifications that the reviewers requested, which improved the consistency and quality of the manuscript. It can be accepted for publication.

Reviewer #5: Many thanks for the opportunity to review this well-thought out analysis. I suggest that the discussion section could be strengthened by discussing the limitations of the data in not being able to account for underdiagnosis of scabies, especially in resource limited settings, and the lack if ability to measure the impact of secondary infections and their sequelae, such as group A strep. This paper neatly discusses these limitations of the epidemiologic data for scabies: Cox V, Fuller LC, Engelman D, Steer A, Hay RJ. Estimating the global burden of scabies: what else do we need? Br J Dermatol. 2021 Feb;184(2):237-242. doi: 10.1111/bjd.19170. Epub 2020 Jul 9. PMID: 32358799.

Reviewer #6: (No Response)

Reviewer #7: Congratulations to the authors on what seems to be a very comprehensive analysis. Although the public health relevance is definitely there, I found little novelty with the conclusions of the research to the field of scabies epidemiology research. Not to say this research is not worthy of publishing, I do wonder on the scope of its contribution to the sector.

PLOS authors have the option to publish the peer review history of their article (what does this mean?). If published, this will include your full peer review and any attached files.

Reviewer #2: **Yes: **Efrén Murillo-Zamora

Reviewer #4: **Yes: **Diego-Abelardo Alvarez-Hernández

Reviewer #5: No

Reviewer #6: No

Reviewer #7: **Yes: **Dorin Brener Turgeman

---

## [Editor Report · Decision Letter 2]

10 Dec 2024

Dear doctor Li,

We are pleased to inform you that your manuscript 'The disability-adjusted life years (DALYs), prevalence and incidence of scabies, 1990–2021: a systematic analysis from the Global Burden of Disease Study 2021' has been provisionally accepted for publication in PLOS Neglected Tropical Diseases.

Best regards,

Benedikt Ley, PhD

Guest Editor

Jong-Yil Chai

Section Editor

Shaden Kamhawi

co-Editor-in-Chief

Paul Brindley

co-Editor-in-Chief

---

## [Editor Report · Acceptance letter]

17 Dec 2024

Dear doctor Li,

We are delighted to inform you that your manuscript, "The disability-adjusted life years (DALYs), prevalence and incidence of scabies, 1990–2021: a systematic analysis from the Global Burden of Disease Study 2021," has been formally accepted for publication in PLOS Neglected Tropical Diseases.

Best regards,

Shaden Kamhawi

co-Editor-in-Chief

Paul Brindley

co-Editor-in-Chief
